# Digital Mapping of Soil Profile Properties for Precision Agriculture in Developing Countries

Antonio López-Castañeda [1] , Joel Zavala-Cruz [1] , David Jesús Palma-López [1] , Joaquín Alberto Rincón-Ramírez [1] and Francisco Bautista [2,*]

1   Colegio de Postgraduados, Campus Tabasco, Heroica Cárdenas 86500, Tabasco, Mexico; tonolc@colpos.mx (A.L.-C.); zavala_cruz@colpos.mx (J.Z.-C.); dapalma@colpos.mx (D.J.P.-L.); jrincon@colpos.mx (J.A.R.-R.)
2   Centro de Investigación en Geografía Ambiental, Universidad Nacional Autónoma de México, Morelia 58190, Michoacán, Mexico
*   Correspondence: leptosol@ciga.unam.mx

**Abstract:** The soil profile and its spatial distribution are two essential aspects for promoting sustainable agriculture, with precise inputs in quantity, space, and time. This work's objective was to elaborate a digital map of soil fertility considering the complete profile for the accurate management of amendments and fertilizers. For the preparation of the soil fertility map, the following inputs were used: a digital elevation model, information from 44 soil profiles, the conversion of the properties of the soil profiles into surface units, geostatistical analysis of the soil properties, and the preparation of the final map with a geographic information system. The best spatial models were achieved with CEC, pH, Ca, Mg, Na, and K. The map of the soil fertility classes was produced considering CEC and the pH value. The soil fertility classes presented the following sequence of occupied surfaces: very low, medium, very high, high, and low. A process was generated to elaborate digital maps (geostatistics) of soil fertility using taxonomic information from soil profiles and considering the complete profile. The process converts soil classification into geographic and soil fertility information from basic science to application.

**Keywords:** geostatistics; soil mapping; semivariogram; spatial variability

## 1. Introduction

Sustainable development objectives require a type of agriculture oriented toward caring for the environment with the highest possible yields [1,2]. For this, it is necessary to leave behind concepts such as the topsoil without taking into account the complete soil profile; the use of regional fertilization rates based on NPK, without taking into account the diversity of soils; and the application of amendments and other agricultural inputs without a precise diagnosis.

Detailed information on the soil profile and its spatial distribution is essential for promoting sustainable agriculture, with precise inputs in quantity, space, and time [3–7]. In particular, accurate and updated soil attributes allow for better and more efficient fertility management, benefing crop productivity and sustainability [8–10].

In practice, precision agriculture is not being carried out in developing countries due to several reasons, such as the excessive division of land into small plots, the lack of capacity in data generation to produce plot maps, the insufficient technical ability of the producers to make the diagnoses, and the scarcity of laboratories, among others [2,11]. Given this situation, the generation of methodological strategies is required to achieve better diagnoses of soil fertility.

In the state of Tabasco, agriculture is the main productive activity after oil. Productive differences are evident in plains and hills. In the plains, cocoa, sugar cane, corn, beans, and cultivated pasture are grown [12,13], while in the hills, the main crops are citrus, pineapple,

sugar cane, cultivated pasture, and forest plantations of eucalyptus, pine, acacia, and rubber [14,15]. However, there are a variety of soils both in the plains and in the hills; the main groups of soils according to the WRB (2015) are as follows: in the plains Vertisols (VR), Gleysols (GL), Fluvisols (FL), and Technosols (TC); and in the hills are the Acrisols (AC), Cambisols (CM), and Umbrisols (UM) [16,17]. These soil groups present two main agricultural problems, excess moisture (GL, FL, VR), low chemical fertility (AC, UM), low pH values, and low base saturation [17].

The soils and crops are different, but fertilization varies very little, and it is mainly focused on nitrogen, phosphorus, and potassium [13–15,18,19]. The farmers recognize the productive differences between the hills and the plains soils [19]. Still, the knowledge of the relief is insufficient to achieve agriculture by the objectives of sustainable development. It is required to improve the precision of land use.

In recent years, Soil & Environment software was generated to express soil properties in surface units of either square meters or hectares [20–22]. The study of the soil profile for improving agriculture is beyond the so-called arable layer.

Geostatistics is the primary discipline used in studies of the spatial heterogeneity of soils that helped in popularizing precision agriculture or site-specific agriculture. Geostatistics is used to create maps and thus estimate the value of soil properties for unsampled sites [8], leading to cheaper and faster plot mapping [23–25]. One of the main weaknesses of parcel maps produced with geostatistical techniques lies in the use of topsoil (0 to 30 cm deep), arguing that it is there where the nutrition of crops is carried out, which is partially true. However, most of the soils used in agriculture are more than 1 m or deeper; that is, a large amount of fine earth is ignored, which also influences the crops' nutrition and support.

The work's objective was to elaborate a methodological strategy for creating digital maps of soil fertility considering the complete soil profiles in Tabasco, Mexico, in conditions of scarce data, as is often the case in developing countries.

## 2. Materials and Methods

### 2.1. Study Zone

The study zone is in the municipalities of Huimanguillo and Cárdenas, Tabasco. The study area is 29,398 ha, which is located between the geographic coordinates of 17°49′00″ and 18°01′00″ north latitude, and 93°31′30″ and 93°44′30″ west longitude. The climate is Am (f) warm humid with abundant rains in summer, with an annual rainfall of 2400 to 2800 mm; the average annual temperature varies from 24 to 28 °C [26]. The study area is located at an altitude of 0 to 34 masl; it comprises a plain and hills (Figure 1).

The plain is composed of alluvial sediments from the Holocene Quaternary, with slopes of 0 to 6% and relative heights of 1 to 4 m; it is cultivated with sugar cane, cocoa, and grasslands. The hills are composed of sand, silt, sandstone, and conglomerate sediments from the Pliocene–Pleistocene; it has slopes of 1% to 30% and relative heights of 2 to 14 m; pasture, citrus, pineapple, and secondary vegetation are cultivated [27,28].

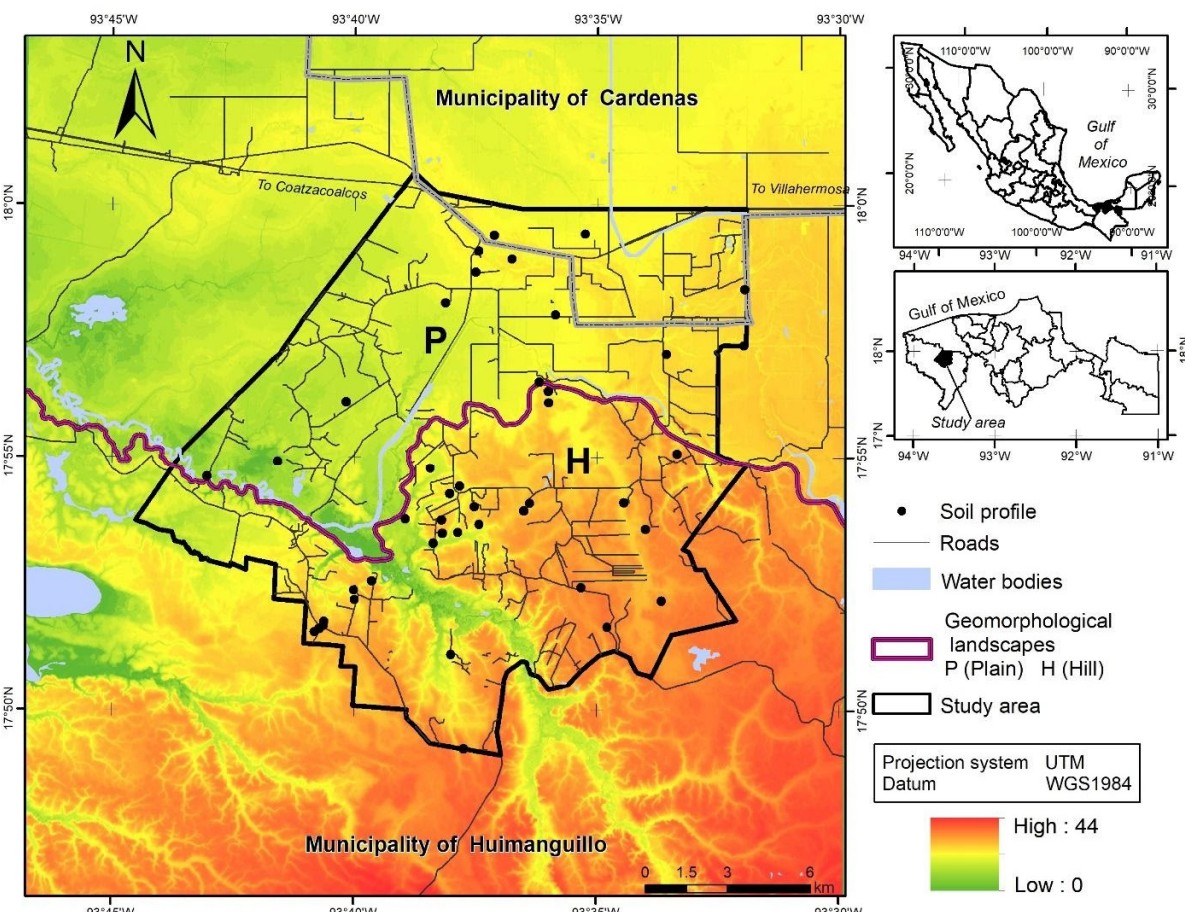

**Figure 1.** Location of study zone.

### 2.2. Soil Sampling and Analysis

We used a legacy database with 19 soil profiles [14–16]. We surveyed 25 soil profiles in the landforms with no soil profiles (Figure 1). Information from each sampling site was recorded, such as altitude, slope, vegetation cover, geographic coordinates, and surface drainage [29]. The soil samples were dried in the shade and sieved with a 2 mm mesh.

The physical and chemical analyses of the soil samples were as follows: pH measured with potassium chloride (1:2.5); pH determined by potentiometry in water (1:2) [30]; electrical conductivity; phosphorous by Olsen method; total nitrogen by the semimicro Kjeldahl method; exchangeable bases (Ca, Mg, K, and Na) and cation exchange capacity (CEC) by extraction with 1 N ammonium acetate, pH 7.0 [31]; soil organic matter (SOM) by the Walkley and Black technique [32]; texture with the Bouyoucos technique [33], coarse fragments (%); and bulk density (BD) by the waxed aggregate method [34].

We used Soil & Environment software to convert units of soil properties into surface units considering the complete profile; this procedure is called soil–ecological evaluation [20–22]. For example, field capacity in L m$^{-2}$; organic matter in kg m$^{-2}$; cation exchange capacity, calcium, magnesium, sodium, and potassium in mol m$^{-2}$. The pH value, nitrogen (%), and phosphorus (mg kg$^{-1}$) were used only for topsoil.

The first step is the calculation of the fine earth using the following equation:

$$FE_i \ (kg \ m^{-2}) = BDi \ (g \ cm^{-3}) * TH_i \ (dm) * (1 - (CR_i \ [Vol.\%]/100))$$

where $FE_i$ (kg m$^{-2}$) is the fine earth and refers to soil particles smaller than 2 mm of soil horizon i in kg m$^{-2}$; $BDi$ (g cm$^{-3}$) is the bulk density of soil horizon i in g cm$^{-3}$; $TH_i$ (dm) is

the thickness of horizon i in decimeters; and $CR_i$ (Vol.%) is the volume of coarse fragments in soil horizon i.

$$FE_t\left(kg\ m^{-2}\right) = \sum_{i=n}^{i=1} BD_i\left[g\ cm^{-3}\right] * TH_i[dm] * \left(1 - \left(\frac{CR_i[\%Vol.]}{100}\right)\right) * 100$$

The content of organic carbon is calculated as

$$SOCi\ (kg\ m^{-2}) = FEi\ (kg\ m^{-2}) * (Ci\ [\%]/100)$$

where SOCi ($kg\ m^{-2}$) is the content of organic carbon at depth i in $kg\ m^{-2}$; FEi ($kg\ m^{-2}$) is fine earth and refers to soil particles smaller than 2 mm at depth i in $kg\ m^{-2}$; and Ci (%) is the carbon percentage at depth i.

Total nitrogen and field capacity units are expressed as a percentage but then converted to $kg\ m^{-2}$ in the same way as SOC.

The amounts per square meter of the exchangeable cations and the CEC were calculated with the same procedure:

$$CEC_t\left(mol\ m^{-2}\right) = \sum_{i=n}^{i=1} FE_i\left[kg\ m^{-2}\right] * CEC_i\left[cmol\ kg^{-1}\right] * 0.01$$

where $CEC_t$ ($mol\ m^{-2}$) is the content of cation exchange capacity at depth i in $mol\ m^{-2}$; FEi ($kg\ m^{-2}$) refers to the soil particles smaller than 2 mm at depth i in $kg\ m^{-2}$; and $CEC_i$ ($cmol\ km^{-1}$) is the cation exchange capacity at depth i.

Database and spatial analyses.

The exchangeable cations (Ca, Mg, Na and K) are converted from $cmol\ kg^{-1}$ to $mol\ m^{-2}$ in the same way as the CEC [20–22].

A georeferenced data matrix based on soil properties was constructed. Subsequently, geostatistical analysis using kriging interpolation was performed with Gamma Design software [35]. First, an exploratory data analysis was performed in order to identify extreme values and to verify the Gaussian distribution of the analyzed parameters. Spatial autocorrelation measures the similarity of a variable within a specific area, and it describes the spatial continuity based on the experimental semivariogram. The experimental semivariogram, defined as the arithmetic mean of all the squares of the differences between pairs of experimental values separated by a given distance, is represented by:

$$\gamma(h) = \frac{1}{2n}\sum_{i=1}^{n}[Z(X_i) - Z(X_i + h)]^2$$

where $\gamma$ (h) is the estimated or "experimental" semivariance value for all pairs at a lag distance h; $Z(X_i)$ is the soil property values in the point i; $Z(X_i + h)$ is the soil property values of other points separated from xi by a discrete distance h; xi are the georeferenced positions where the $Z(X_i)$ values were measured; and n represents the number of pairs of observations separated by a distance h [34,35].

The structural parameters of the semivariogram that describe the model are (a) the variance of the nugget (Co), which is the y-intercept of the semivariogram model that represents the variation in the analyzed parameter that does not depend spatially on the interval examined and which reflects both spatial variations at distances shorter than the minimum sampling space and unexplained measured parameter variation; (b) the threshold (Co + C), which indicates the asymptote of the curve where the structural variance reaches its maximum values because it remains constant, and (c) the range indicates the value of the distance (degree) in which the maximum variance of the studied parameter is reached, which defines the area of influence of the autocorrelation. Next, the theoretical model that best fits the experimental semivariogram is selected [24,36].

The estimation of the data was done using kriging interpolation, which is a technique that provides the best unbiased linear estimator as well as an estimation error known

as the kriging variance, which depends on the correlation structure chosen, based on the theoretical model and the locations of the original data. The interpolation attributes weight to each observed value while considering the geometric characteristics of the data. Minimizing the estimation variance guarantees the optimal use of the available information [37]. Soil property estimates were obtained by punctual kriging using the following equation:

$$Z(Xo) = \sum_{i=1}^{n} \lambda i \, Z(X_i)$$

where $\lambda i_i$ is the optimal weights selected to minimize the estimation variance, $Z(X_i)$ is the observed value of a soil property, and $Z(X_0)$ is the optimal and unbiased estimate of a soil property.

Maps were edited with ArcGIS 10.1 software using the UTM projection, zone 15N of Datum WGS1984 [38].

## 3. Results

pH values ranged from acidic to neutral, and CEC, Ca, and Mg values presented more significant variation coefficients than 82%. Exchangeable monovalent cations showed low amounts of less than 2.26 mol m$^{-2}$, with high standard deviations concerning the mean values, generating high variation coefficients (Table 1).

The pH value was moderately acidic in the plain, which coincides with Vertisols, Fluvisols, and Cambisols [14,15]. Unlike the hills, soils of the plains showed more significant differences among each other because greater amounts of calcium and potassium were observed in the northeast but not in the west.

In the northeast are located the most fertile soils in the plains with higher amounts of CEC, Ca, and K, and soils with less acidity; however, they are those with the lowest organic matter content, which is a situation that may be due to agricultural management with sugar cane plantations.

The K map is essential in the fertilization of crops because it is one of the main macronutrients that farmers use. However, the K map should be used with caution because the parameters of the spatial model are of the lowest quality due to the high random variability indicated by the value of the nugget.

The experimental semivariograms of all variables except Nt and SOC fit the Gaussian model (Table 2).

The models explain the spatial variation because the nugget values were low (random variance) except for K, Nt, SOC, and EC (Table 2).

**Table 1.** Descriptive statistics of soil properties expressed per unit area.

| Parameters | Mean | Skewness | Kurtosis | Minimal Value | Maximal Value | Standard Deviation |
|---|---|---|---|---|---|---|
| pH | 4.99 | 0.93 | −0.30 | 3.83 | 7.33 | 0.9 |
| EC (dS m$^{-1}$) | 0.047 | 1.45 | 2.33 | 0.00 | 0.24 | 0.05 |
| COS (t ha$^{-1}$) | 129.9 | 1.20 | 2.16 | 43.07 | 330.13 | 56.8 |
| P (mg kg$^{-1}$) | 12.32 | 2.79 | 9.48 | 0.00 | 74.54 | 26.9 |
| Nt (%) | 0.175 | 1.07 | 2.97 | 0.01 | 0.58 | 0.11 |
| CEC (mol m$^{-2}$) | 173.2 | 1.18 | 0.17 | 22.22 | 570.12 | 143.1 |
| Ca (mol m$^{-2}$) | 78.4 | 1.41 | 0.83 | 0.40 | 407.67 | 112.8 |
| Mg (mol m$^{-2}$) | 38.3 | 1.42 | 0.60 | 0.41 | 184.41 | 56.1 |
| K (mol m$^{-2}$) | 1.4 | 0.86 | 0.11 | 0.02 | 4.65 | 1.1 |
| Na (mol m$^{-2}$) | 2.3 | 3.04 | 11.45 | 0.01 | 18.36 | 3.3 |
| FC (L m$^{-2}$) | 617.3 | −2.45 | 6.65 | 239.48 | 815.10 | 1.0 |

The adjustment of the experimental semivariogram to the theoretical semivariogram had the following sequence: Ca > Mg > Na > pH > CEC > K > FC (Table 2).

Good spatial models were not achieved with some soil properties due to either the non-Gaussian distribution (P) or the high values of the nugget (K, Nt, SOC, and EC), and some cases generated low levels of adjustment to the theoretical model (Nt, SOC, and FC). For these reasons, the interpolations of these soil properties were not carried out.

When comparing the maps with the border between the plain and hills, the best identified maps were CEC, pH, Mg, Ca, and Na (Table 2).

The highest concentrations of CEC (200 to 400 mol m$^{-2}$) were located to the north and northwest in the plain. On the other hand, low CEC values (100–200 mol m$^{-2}$) were located in the center and, to the south, in the hills. The CEC made it possible to identify the transition gradient right on the border between the plains and hills (Figure 2).

The pH value was less than 5 in the hills and greater than 5 in the plains. The pH value also allowed for identifying the transition between the plain and hills (Figure 2).

The highest values of interchangeable Mg (100 to 150 mol m$^{-2}$), Ca (100 to 200 mol m$^{-2}$), and Na (3 to 5 mol m$^{-2}$) are located to the north in the plain. On the other hand, the low values were situated to the south, 10, 20, and 2 mol m$^{-2}$, respectively, in the hills (Figure 3).

**Table 2.** Characteristics of semivariogram models of soil properties.

| Soil Properties | Model | Nugget | Sill | Range (m) | Model r$^2$ |
|---|---|---|---|---|---|
| CEC (pH covariant) | Gaussian | 0.10 | 92 | 5520 | 0.80 |
| CEC (mol m$^{-2}$) | Gaussian | 0.12 | 2.22 | 13590 | 0.86 |
| pH | Gaussian | 0.13 | 1.02 | 5530 | 0.87 |
| Ca (mol m$^{-2}$) | Gaussian | 0.31 | 3.63 | 4160 | 0.96 |
| Mg (mol m$^{-2}$) | Gaussian | 0.71 | 3.98 | 6310 | 0.90 |
| Na (mol m$^{-2}$) | Gaussian | 0.01 | 18.54 | 2963 | 0.88 |
| K (mol m$^{-2}$) | Gaussian | 0.67 | 2.02 | 5470 | 0.65 |
| P (mg kg$^{-1}$) | Gaussian | 2.60 | 149.10 | 27.60 | 0.86 |
| EC (dS m$^{-1}$) | Gaussian | 0.01 | 0.03 | 5890 | 0.80 |
| Nt (%) | Exponential | 0.01 | 0.04 | 30780 | 0.52 |
| SOC (t ha$^{-1}$) | Linear | 0.18 | 0.18 | 13498 | 0.48 |
| FC (L m$^{-2}$) | Gaussian | 0.03 | 0.15 | 7787 | 0.30 |

The highest amounts of exchangeable K ($\geq$2 mol m$^{-2}$) were located to the north and northwest, in the plain; however, in the same plain, to the east and the center, the values were $\leq$2 mol m$^{-2}$. In the hills, the K contents were highly variable, but the zone with the lowest values was to the south (<0.55 mol m$^{-2}$) (Figure 3). The exchangeable Na presented low amounts in the study area, which were slightly higher in the plains compared to the hills (Figure 3).

Cross-validation of the interpolations for the pH and CEC values indicated that the parameters had r2 coefficients of 0.86 and 0.47, and standard error values of 0.056 and 0.23, respectively (Figure 4). SE values suggest that the size of the error obtained with interpolation models is low in pH and CEC. That is, the interpolation models were acceptable for mapping pH at detail scales (1:50,000) and scarily sufficient for CEC.

In the study area, five classes of soil fertility were found (Figure 5); toward the north was the class of very high fertility within the plain, with high values of cation exchange capacity, slightly acidic pH values, and high values of exchangeable bases (Figures 2 and 3). The high soil fertility class four surrounds the five very high soil fertility class zones.

To the center and east to west is the zone of medium soil fertility, right in the transitional zone between the plains and hills (Figure 5).

In the center and to the south in the hills, the surface of very low fertility (Acrisols) is located, with the most acidic pH values and shallow CEC values. The two low fertility class areas are situated to the south.

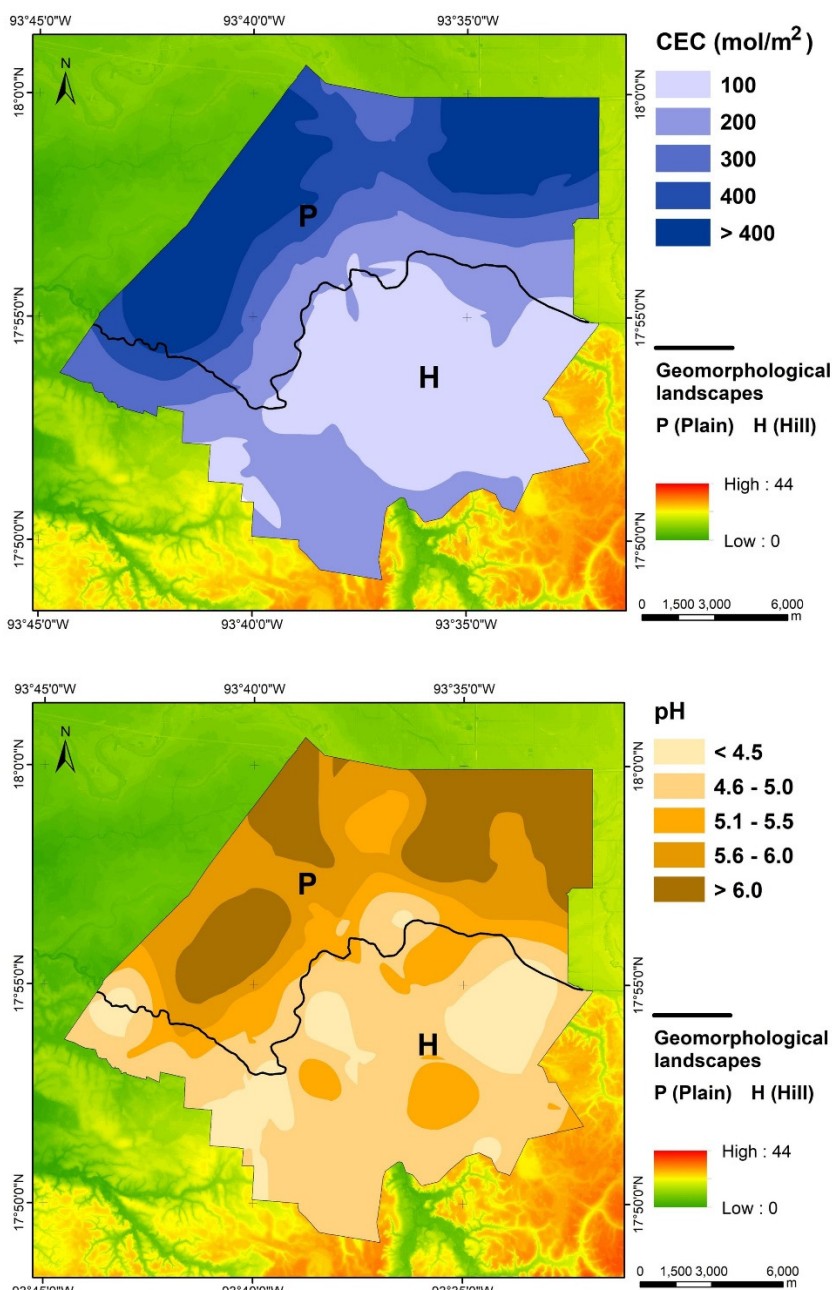

**Figure 2.** Spatial distribution of soil properties CEC and pH.

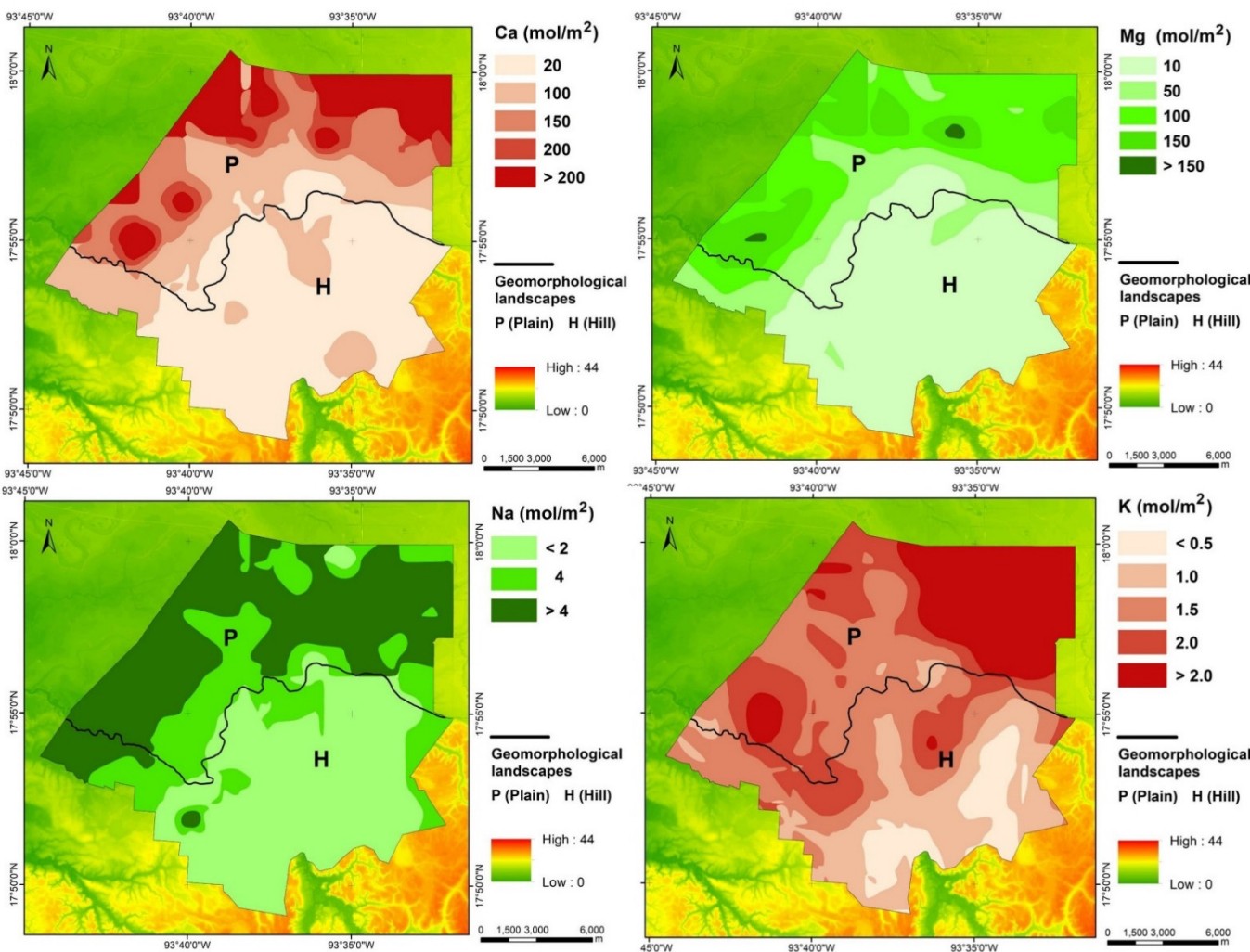

**Figure 3.** Spatial distribution of exchangeable cations.

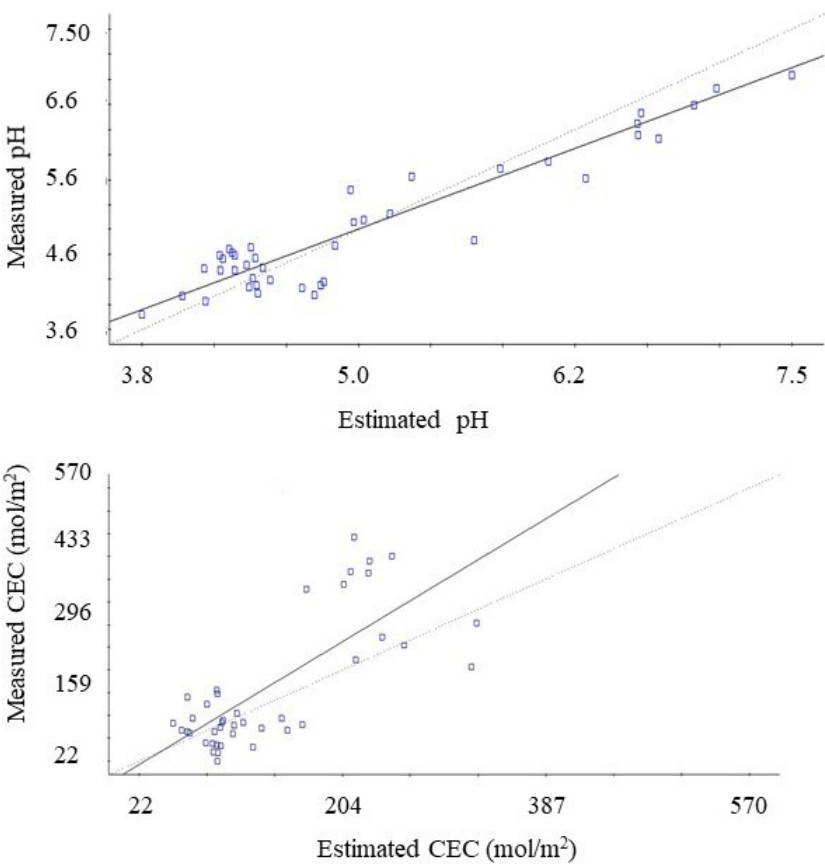

**Figure 4.** Cross-validation graphs of selected soil properties (pH and CEC).

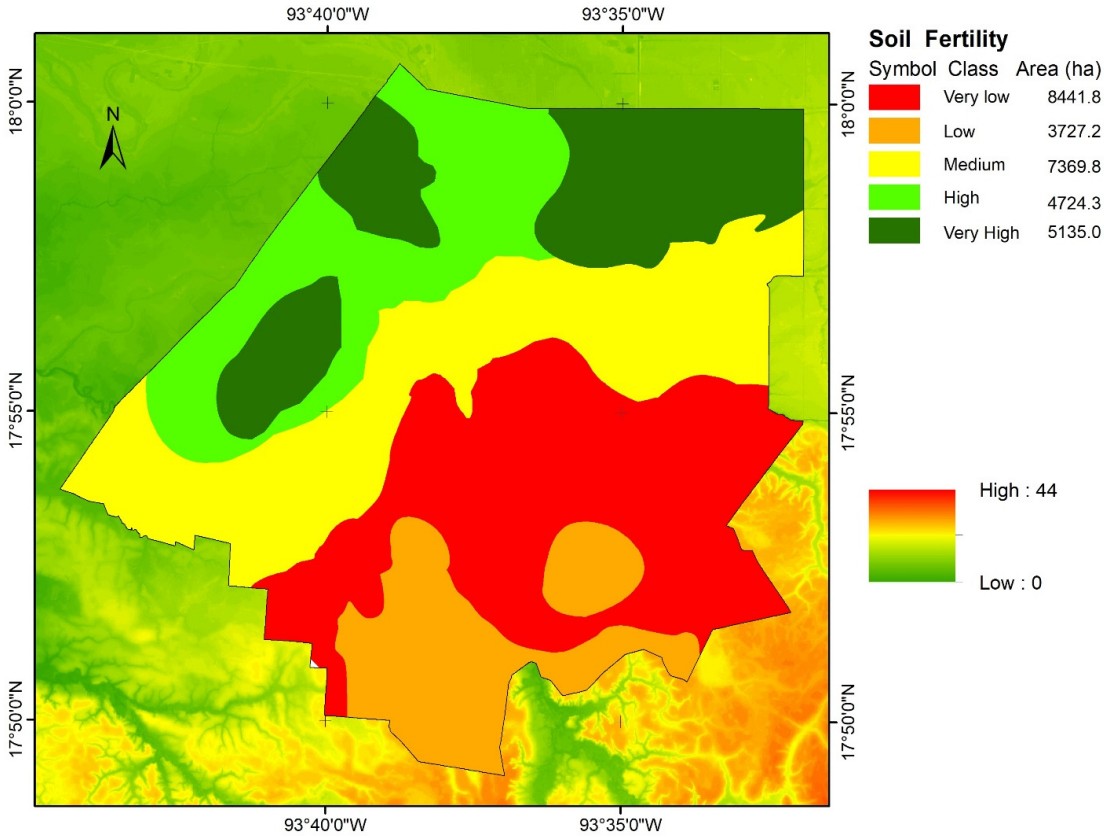

**Figure 5.** Model of soil fertility map.

## 4. Discussion

### 4.1. Soils, Fertility, and Management

The very acidic pH values of the soils of the hills correspond to Acrisols [15,17]. In these plots, the application of amendments with dolomite is recommended to add calcium and magnesium. Another option is the application of dissolved organic matter-type vinasse that also contains high concentrations of K, Ca, Mg, and other nutrients [39]. The application of nixtamalization wastewater is also recommended because it contains high calcium concentrations.

The objectives of applying mineral, organic, or mixed amendments must be to increase the pH, decrease the exchangeable aluminum, and increase the content and availability of nutrients such as Ca, Mg, and K, focusing on the balance among them [13,15].

The elaboration of maps considering the complete soil profile is not joint in traditional agricultural studies or precision agriculture studies. Figure 6 shows five soil profiles of the study area in which the arable layer is indicated with a yellow line; everything below the yellow line is left out of conventional studies. The soil-ecological evaluation considers two significant advantages: (a) the transformation of units when passing from concentration to quantity (units per surface) per horizon; (b) the use of a complete soil profile [20–22,40]. This procedure facilitates the calculation of the amounts of fertilizers, manures, and amendments in kg m$^{-2}$ or t ha$^{-1}$. However, it would be necessary to consider the other properties of the soil profile, for example, physical and chemical restrictions to specific crops (types of internal drainage, compaction of horizons, toxicity, salinity, etc.), as recommended by the fertility capability classification (FCC) method [40].

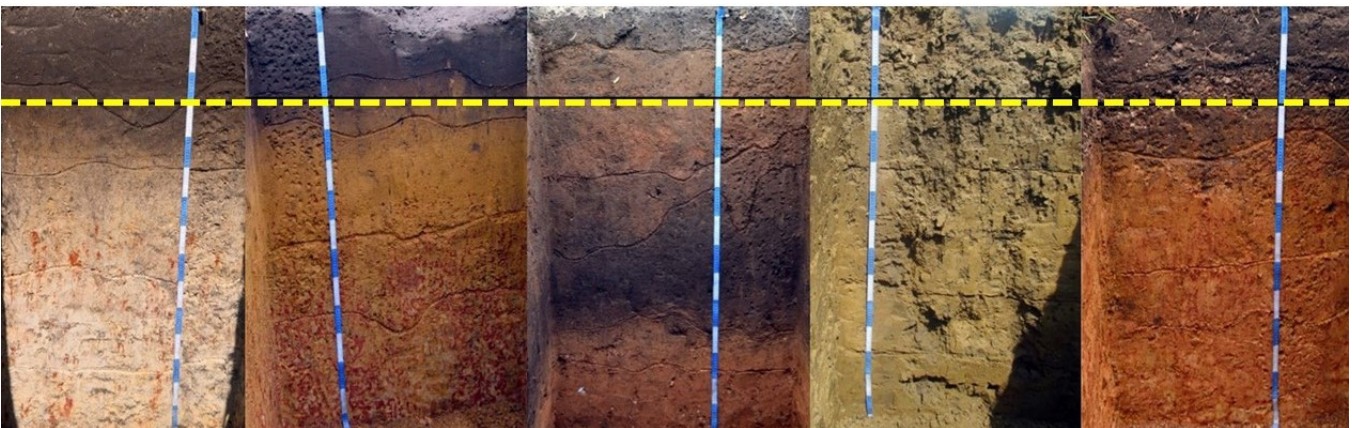

**Figure 6.** Photographs of five soil profiles of the study area indicating the topsoil (30 cm). The horizontal-yellow line indicates the "arable layer of the soil"; The vertical line with alternating colors every 10 cm is the tape measure.

### 4.2. Method

Conventional soil mapping techniques use the relief to elaborate soil maps (considering the complete soil profile) at various scales [5]. These soil maps are often coarse in scale and without detail, not to say inaccurate in spatial boundaries and attribute data [10]. In this work, we added digital soil mapping techniques to identify transitions between unsampled surfaces. That is, we used a hybrid approach. In the study area, farmers use the relief (plains and hills) as indicators of spatial heterogeneity [15–17,41]. Now, with the results of this study, they have two more parameters, the pH value and the CEC. However, to move to the plot level, it will be necessary to identify other soil properties to improve the maps' accuracy.

In geography, maps at a scale of 1:50,000 are considered detailed scale maps; they are of agricultural use in developing countries, even assuming that the idea is to reach the level of parcel maps, especially in areas of high spatial heterogeneity such as in the state

of Tabasco [12–19,41]. Soil maps at a detailed scale of 1:50,000 are disqualified because a map represents a large surface; however, when the accuracy of the models is acceptable, the maps are usually handy [24,37,42,43].

The spatialization of soil properties with geostatistical models had the advantage of using 44 soil profiles (only 25 are new) to prepare a map of 29,398 ha, that is, a map with a rate of 659 ha per soil profile. Another advantage was not needing to classify soil profiles given the limited number of professionals in this discipline. The method proposed in this study for areas of thousands of hectares and with fewer than 50 soil profiles may be of great interest to developing countries. Geostatistical models were good, even though sampling points were fewer than 100 and not homogeneously distributed. It is recommended that in the application of this method, the sampling be systematic [25,36].

### 5. Conclusions

The procedure for the preparation of low-cost soil fertility maps considered: (a) a legacy taxonomic soil database; (b) a survey of 19 soil profiles; (c) the conversion of the databases to units of soil properties per surface units ($mol/m^2$ or mol/ha) using complete soil profile properties; (d) a geostatistical analysis,

The digital elevation model allowed identifying two geomorphological landscapes: plains and hills. Geostatistical models showed the properties of the soils of both geomorphological landscapes and their variants.

Both the calcium and pH maps allowed the identification of the differences within the soils of the plain.

The digital soil map can be helpful for more specific agronomic management recommendations based on the requirements of the crops in the study area, that is, precision agriculture.

**Author Contributions:** Conceptualization, F.B. and J.Z.-C. Methodology, F.B. and A.L.-C. Validation, J.Z.-C. Formal Analysis, F.B. and J.A.R.-R. Investigation, A.L.-C., J.Z.-C. and D.J.P.-L.; Data Curation, F.B.; Writing—Original Draft Preparation, F.B., J.Z.-C. and A.L.-C. Writing—Review and Editing, F.B. Visualization, D.J.P.-L. Supervision, D.J.P.-L., J.Z.-C. and J.A.R.-R. Project Administration, J.Z.-C. All authors have read and agreed to the published version of the manuscript.

**Funding:** To the COLPOS: Campus Tabasco.

**Institutional Review Board Statement:** Not applicable.

**Informed Consent Statement:** Not applicable.

**Acknowledgments:** To the CONACYT for the scholarship 207738 granted to the first author to complete doctoral studies. To the LASPA laboratory of the Campus Tabasco, for the facilities during the physical and chemical analyzes. To the COLPOS, Campus Tabasco for logistical support during the study. To the Act with Science Company for the loan of the Soil & Environment software license.

**Conflicts of Interest:** The authors declare no conflict of interest.

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
