# Peer review of "Digital Mapping of Soil Profile Properties for Precision Agriculture in Developing Countries"

_agronomy, doi:10.3390/agronomy12020353_

Round 1
Reviewer 1 Report
In this paper with the title of "Digital mapping of soil profile properties for precision agriculture in developing countries", the aimed to elaborate a digital map of soil fertility considering the complete soil profiles for the precise management of amendments and fertilizers, which can be useful for more specific agronomic management recommendations based on the requirements of the crops in the study area, that is, precision agriculture. The authors did this job nicely by providing sufficient evidences and the research in this manuscript is important and relevant to the scope of the journal. I recommend to publish this work after addressing some issues as listed bolow:
(1) Line 39-48: The authors should present any relevant previous studies that have been done in the study area (Tabasco, Mexico).
(2) L63-99: A section on "Soil sampling and analysis", more details regarding the statistical analysis need to be mentioned.
(3) L179-194: Compare the results in this study with those in previous studies.
(4) L157: Table 3 is perhaps better represented as a picture.
(5) What specific guiding significance does this study provide procedure for the elaboration of soil fertility maps, used a digital elevation model, or taxonomic information of the soil? Please make further statements in the discussion.
Author Response
(1) Line 39-48: The authors should present any relevant previous studies that have been done in the study area (Tabasco, Mexico).
Okay, we modify the introduction and we introduce several references of very important articles carried out in the study area:
Salgado-García, S.; Palma-López, D.J.; Zavala-Cruz, J.; Lagunes-Espinosa, L.C.; Castelán-Estrada, M.; Ortiz-García, C.F. Sistema Integrado Para Recomendar Dosis de Fertilizantes En Caña de Azúcar (SIRDF): Ingenio Presidente Benito Juárez; Colegio de Postgraduados, Campus Tabasco: H. Cárdenas, Tabasco. México, 2013
Aguilar-Rodríguez, J.R.; Zavala-Cruz, J.; Juárez-López, F.; Palma-López, D.J.; Castillo-Acosta, O.; Shirma-Torres, E. Aptitud Edáfica de Eucalyptus Urophylla ST Blake En La Terraza de Huimanguillo, Tabasco, México. AGROProductividad 2017, 10, 79–85.
Tinal-Ortiz, S.; López, D.J.P.; Zavala-Cruz, J.; Salgado-García, S.; Palma-Cancino, D.J.; Hidalgo-Moreno, C.I. Degradación química en Acrisoles bajo diferentes usos y pendientes en la sabana de Huimanguillo, Tabasco, México. Agro Product. 2020, 13, doi:10.32854/agrop.vi.1603.
Zavala Cruz, J.; Salgado García, S.; Marín Aguilar, Á.; Palma López, D.J.; Castelán Estrada, M.; Ramos Reyes, R. Transecto de suelos en terrazas con plantaciones de cítricos en Tabasco. Ecosistemas Recur. Agropecu. 2014, 1, 123–137.
López-Castañeda, A.; Zavala-Cruz, J.; Palma-López, D.J.; Bautista-Zuñiga, F.; Rincón-Ramírez, J.A. Formas Del Terreno a Escala Detallada En Planicies y Lomeríos Del Municipio de Huimanguillo, Tabasco, México. Bol. Soc. Geológica Mex. 2022, (en prensa).
Salgado, S.; Palma, D.J.; Zavala, J.; Lagunés, L.C.; Mepivosteh, C.E.; Ortiz-García, C.F.; Júarez-López, J.F.; Ruiz-Rosado, O.; Armidia-Alcudia, L.; Rincon-Ramírez, J.A. Sistema integrado para recomendar dosis de fertilizantes en caña de azúcar (SIRDF): Ingenio Presidente Benito Juárez; Colegio de Postgraduados-Campus Tabasco: H. Cárdenas, Tabasco, México, 2013; ISBN 978-607-7533-18-4.
Salgado-García, S.; Palma-López, D.J.; Zavala-Cruz, J.; Ortiz García, C.F.; Castelán-Estrada, M.; Lagunes-Espinoza, L.C.; ., Guerrero-Peña, A.; Ortiz-Ceballos, A.L.; Córdova-Sánchez, S. Sistema Integrado Para Recomendar Dosis de Fertilizantes (SIRDF): En La Zona Piñera de Huimanguillo, Tabasco; Colegio de Postgraduados: H. Cárdenas, México., 2010;
Salgado García, S.; Palma López, D.J.; Zavala Cruz, J.; Ortiz García, C.F.; Lagunés Espinoza, L. del C.; Castelán Estrada, M.; Guerrero Peña, A.; Ortiz Ceballos, Á.I.; Córdova Sánchez, S. Integrated System for Recommending Fertilization Rates in Pineapple (Ananas Comosus (L.) Merr.) Crop. Acta Agronómica 2017, 66, 566–573, doi:10.15446/acag.v66n4.62257.
Salgado-García, S.; Colorado, J.A.; Salgado-Velázquez, S.; Sánchez, S.C.; López, D.P.; Ramírez, J.A.R.; Castañeda, A.L. Spatial Variability of Some Chemical Properties of a Cambisol Soil with Cocoa (Theobroma Cacao L.) Cultivation: Variabilidad Espacial. Agro Product. 2021, 14, doi:https://doi.org/10.32854/agrop.v14i1.1706.
(2) L63-99: A section on "Soil sampling and analysis", more details regarding the statistical analysis need to be mentioned.
Okay, we put in the details of converting soil properties from concentration units to area units. We also put the details of the geostatistical analysis.
(3) L179-194: Compare the results in this study with those in previous studies.
Okay, we rewrite the discussion. Specifically at this point we wrote this in the discussion:
Conventional soil mapping techniques use the relief to elaborate soil maps (considering the complete soil profile) at various scales [5]. These soil maps are often coarse in scale and without detail, not to say inaccurate in spatial boundaries and attribute data [10]. In this work, we added digital soil mapping techniques to identify transitions between unsampled surfaces. That is, we used a hybrid approach. In the study area, farmers use the relief (plains and hills) as indicators of spatial heterogeneity [15-17, 42]. Now, with the results of this study, they have two more parameters, the pH value and the CEC. However, to move to the plot level, it will be necessary to identify other soil properties to improve the maps' accuracy.
In geography, maps at a scale of 1:50,000 are considered detailed scale maps; they are of agricultural use in developing countries, even assuming that the idea is to reach the level of parcel maps, especially in areas of high spatial heterogeneity such as in the state of Tabasco [12-20, 42]. Soil maps at a detailed scale of 1:50,000 are disqualified because a map represents a large surface; however, when the accuracy of the models is acceptable, the maps are usually handy [25, 38, 43-44].
(4) L157: Table 3 is perhaps better represented as a picture.
Ok, we put the surface data of the classes in the map legend
(5) What specific guiding significance does this study provide a procedure for the elaboration of soil fertility maps, using a digital elevation model, or taxonomic information of the soil? Please make further statements in the discussion.
As we mentioned in the introduction, the sustainable development goals proposed by the UN will not be achieved without precision agriculture throughout the world, however, this desire will not be a reality in developing countries if low-cost procedures are not found that small farmers use them.
In this paper, we report the use of legacy databases of soil taxonomic maps, and basic cartographic processing (ordinary kriging) to produce maps of soil fertility. In addition, we use the complete profile, beyond the topsoil commonly used, but which we know does not include more than two-thirds of the soil (30 cm of a profile more than 100 cm deep).
Reviewer 2 Report
Dear authors,
In my opinion, the work does not have a scientific overtone, it is rather a kind of inventory, as the authors themselves state in line 47: "This work's objective was to elaborate a digital map of soil ...". A scientific publication must have a scientific/cognitive purpose.
In the discussion, the authors state that they have used data from 44 soil profiles to prepare a map covering an area of 29,389 ha (659 ha per soil profile) - such an inference in my view is not supported by any evidence that it can be applied. It may be that this is the right thing to do, but there is a lack of presentation of verification of such reasoning with concrete examples.
Also, the introduction needs to be reworded so that it does not start with Detail information ...
Author Response
We have modified the introduction to expand the topic's context.
We have modified the objective: The work aimed to elaborate a methodological strategy for creating digital maps of soil fertility considering the complete soil profiles in Tabasco, Mexico, in conditions of scarce data, as is often the case in developing countries.
We have introduced a figure with the cross-validation of the soil properties selected to make the map, with which we validate the study.
We have put the details into the methodology, augmented the results, and rewrote 50% of the discussion. We also rewrote the conclusions.
In recent years, a large number of methods have been generated for the preparation of digital soil maps and thus make better use of the soil in all areas, agricultural, livestock, forestry and urban. However, the adoption of these new methods, procedures and/or approaches have not been adopted in many developing countries due to the need for data to make the models and technological capabilities. It is necessary to generate procedures at various scales using pre-existing data so that small farmers have the possibility of moving towards precision agriculture. This is the important point of our study.
Another novel aspect is to use the databases inherited from the taxonomic maps to reduce the number of new soil surveys.
An important point is the use of the soil properties of the complete profile to elaborate fertility maps, these studies are very scarce worldwide.
Round 2
Reviewer 2 Report
No comments